# Knowledge, attitudes, and practices towards childhood tuberculosis among healthcare workers at two primary health facilities in Lusaka, Zambia

**Paul Chabala Kaumba** [1]*, **Daniel Siameka**[1], **Mary Kagujje**[1], **Chalilwe Chungu**[2], **Sarah Nyangu**[1], **Nsala Sanjase**[1], **Minyoi Mubita Maimbolwa**[1], **Brian Shuma**[1], **Lophina Chilukutu**[1], **Monde Muyoyeta**[1]

**1** Centre of Infectious Disease Research in Zambia (CIDRZ), Mass Media, Lusaka, Zambia, **2** Catholic Relief Services, Ibex, Lusaka

* paul.kaumba2@gmail.com

## Abstract

### Background

Zambia is among the 30 high-burden countries for tuberculosis (TB), Human Immunodeficiency Virus (HIV)-associated TB, and multi-drug resistant/rifampicin resistant TB with over 5000 children developing TB every year. However, at least 32% of the estimated children remain undiagnosed. We assessed healthcare workers' (HCWs) knowledge, attitudes, and practices (KAP) towards childhood TB and the factors associated with good KAP towards childhood TB.

### Methods

Data was collected at two primary healthcare facilities in Lusaka, Zambia from July to August 2020. Structured questionnaires were administered to HCWs that were selected through stratified random sampling. Descriptive analysis was done to determine KAP. A maximum knowledge, attitude, and practice scores for a participant were 44, 10, and 8 points respectively. The categorization as either "poor" or "good" KAP was determined based on the mean/ median. Logistic regression analysis was performed to assess the associations between participant characteristics and KAP at statistically significant level of 0.05%.

### Results

Among the 237 respondents, majority were under 30 years old (63.7%) and were female (72.6%). Half of the participants (50.6%) were from the outpatient department (OPD) and antiretroviral therapy (ART) clinic, 109 (46.0) had been working at the facility for less than 1 year, 134 (56.5%) reported no previous training in TB. The median/mean KAP scores were 28 (IQR 24.0–31.0), 7 (IQR = 6.0–8.0) and 5 points (SD = 1.9) respectively. Of the participants, 43.5% (103/237) had good knowledge, 48.1% (114/237) had a good attitude, and

**Data Availability Statement:** Data are available in a code book and correct responses, contained within the Supporting Information files.

**Funding:** Dr. Monde Muyoyeta received the a grant through the Stop TB Partnership/ TB REACH mechanism with funding support from the Government of Canada (TB REACH wave 7) which facilitated the collection of the data for this study. This funding was obtained under The Centre for Infectious Diseases Research in Zambia (CIDRZ). Grant Number: STBP/TBREACH/GSA/W7-7426 Funders site: https://stoptb.org/global/awards/tbreach/wave7.asp The funders had no role in study design, data collection and analysis, decision to publish, or preparation of the manuscript.

**Competing interests:** The authors have declared that no competing interests exist.

54.4% (129/237) had good practice scores on childhood TB. In the multivariate analysis, clinical officers and individuals with 1–5 years' work experience at the facility had higher odds, 2.61 (95% CI = 1.18–5.80, p = 0.018) and 3.09 (95% CI = 1.69–5.65, p = 0.001) of having good attitude respectively, and medical doctors had 0.17 lower odds (95% CI = 0.18–5.80, p = 0.018) of good childhood TB practice. Other participant characteristics didn't show a significant association with the scores.

## Conclusion

The study found suboptimal levels of knowledge, attitude, and practices regarding childhood TB among HCWs. Targeted programmatic support needs to be provided to address the above gaps.

## Introduction

Globally, an estimated 10.6 million people developed tuberculosis(TB) in 2022 of which about 12% were children [1]. Of these children, only 46% were diagnosed and started on treatment [1]. In Zambia, one of the 30 high-burden countries for Tuberculosis (TB), Human Immuno-deficiency Virus (HIV)-associated TB, and multi-drug resistant/rifampicin resistant TB, only 68% of the estimated children with TB were diagnosed and started on treatment in 2022 [1]. Between 2020 and 2021, this proportion did not exceed 50% in either year [2–4].

Diagnosing TB in children can be difficult due to the pauci-bacillary nature of TB in the age group and its variable presentation in children [5–7]. This is compounded by sub-optimal to variable knowledge, attitudes, and practices (KAP) concerning childhood TB among health-care professionals [8–10]. In some settings, healthcare workers (HCWs) hold stigmatizing beliefs and have misconceptions about TB diagnosis, treatment, and infection prevention [8]. Children with symptoms of TB commonly present to primary healthcare facilities [11,12] but due to limited capacity to recognize presumptive TB in children and due to the complexities of diagnosing childhood TB, they are often referred from primary health facilities to referral hospitals and specialized pediatric units resulting in delayed diagnosis [11,12]. Optimizing the knowledge, attitudes, and practices (KAP) of healthcare workers, especially at primary health-care level, is critical to early diagnosis and treatment of childhood TB.

To the best of our knowledge, there is currently no local literature on KAP among health workers towards childhood TB in Zambia. We undertook a study to assess healthcare workers' (HCWs) knowledge, attitudes, and practices towards childhood TB and to determine the factors associated with good knowledge, attitudes, and practices towards childhood TB.

## Methods

### Study design

We conducted a cross-sectional prospective study which was nested within an the main TB REACH Wave 7 study whose objective was to strengthen case finding in children [13].

### Setting

The study was conducted between July 2020 to August 2020 at two primary healthcare facilities, Kanyama and Chawama first level hospitals, that have since been upgraded to secondary

healthcare facilities. In 2019, Kanyama and Chawama hospitals had annual TB notification rates of 814 per 100,000 and 663 per 100,000 respectively. However, only 6% and 4% of their respective total notifications are children.

## Population, sampling and sample size

Our targeted population was registered healthcare workers (nurse, clinical officer and medical officer) supporting departments that should routinely provide TB screening and/or treatment, specifically outpatient department (OPD), Inpatient department (IPD), Antiretroviral therapy (ART) clinic, Maternal Child Health (MCH) clinic and TB clinic.

Stratified random sampling was done; the sample size was calculated using OpenEpi calculator for descriptive studies [14] and the population size in the human resource for health database in 2019. At Kanyama hospital, from a population of 90 nurses, 17 clinical officers and 5 medical doctors, our sample size was 74 nurses, 17 clinical officers and 5 doctors while at Chawama hospital, from a population of 140 nurses, 15 clinical officers and 8 medical doctors we sampled 103 nurses, 15 clinical officers and 8 medical doctors. The overall sample size was 222.

## Data collection and study tools

A paper-based structured questionnaire with 36 questions was used. The questions were adapted primarily from the national pre-training test on childhood TB and the World Health Organisation(WHO) guide to developing KAP surveys [15]. The questionnaire contained five sections including: Demographics (8 questions), Knowledge (18 questions), Attitudes (6 questions) and Practices (4 questions). While some questions required a single response, other questions required multiple responses. All the questionnaires were self-administered by the healthcare workers.

De-identified participant data was entered into the DHIS 2-based electronic database.

## Validation of correct responses

Three medical officers, all trainers on the consolidated training package on TB, independently responded to the KAP questionnaire. Agreement by at least 2 of them was considered as a correct response.

## Statistical analysis

Data was analysed using STATA Statistical Software (Stata Corporation Version 14. College Station, Texas 77845, USA). Descriptive analysis was done to describe the population and summarize the responses. Participant characteristics were reported as frequencies and percentages.

All knowledge, attitude and practice questions were assigned a scoring system wherein correct responses were awarded 1 point, while incorrect responses received 0 points. Responses categorized as "don't know" or "not sure" or "other" were treated neutrally, with a score of 0 points assigned. A total knowledge, attitude, and practice score for each participant was calculated based on the total correct response of 44, 10, and 8 points respectively. The categorization as either "poor" or "good" knowledge, attitude, and practices was determined based on the mean for normally distributed data and median for data that did not exhibit a normal distribution.

Logistic regression analysis was done to investigate the association between characteristics of study participants and knowledge, attitude, and practice. Univariable analysis was initially applied to identify characteristics associated with knowledge, attitude, and practice.

Subsequently, multivariable analysis was conducted, accounting for potential confounding variables. A statistical cutoff point of 0.2 (20%) was used to select variables for inclusion in the multivariable analysis.

## Ethical considerations

Approval for study implementation was obtained from the University of Zambia Biomedical Research Ethical Committee (UNZABREC), IRB Reference number 635–2020 under the TB REACH wave 7 study. The respondents provided written informed consent before being allowed to complete the paper based self-administered questionnaire. The survey protected the confidentiality of the respondents by maintaining anonymous responses.

## Results

### Participant demographics

A total of 237 HCWs participated in the survey (Table 1). Majority of the participants were under 30 years of age, females and nurses at 63.7%, 72.6%,70.5% respectively. HCWs from the outpatient department (OPD) and inpatient department (IPD) accounted for 70.4% of the participants. About half of the participants had been at their respective health facilities for less than a year and more than half of them had not received any training in TB. Among those who had been trained, 47% had been trained on adult TB, 35% had been trained on childhood TB and 43% were trained less than 6 months from the date of the survey.

The median knowledge score was 28 (Interquartile range [IQR] 24.0–31.0) points (Table 2). Overall, 43.5% of respondents had good knowledge. Most respondents (89.0%) correctly identified Zambia as a high burden TB country and 99.6% correctly associated coughing with TB transmission. However, knowledge gaps were observed for other modes of transmission. All participants knew that TB affects the lungs but there was variable knowledge on if it affects other parts of the body. About 49.8% of the participants were aware that extra-pulmonary TB is prevalent in children. The majority didn't correctly distinguish the risk factors for TB infection apart from those for TB disease. At least 88% of participants correctly identified all TB symptoms in children, and about half (48.5%) of the participants knew that Gene Xpert/ Xpert ultra is the first-line TB diagnostic test in Zambia. Less than half of the respondents knew that only one spot sample was required for Xpert testing but 90.5% of the participants knew that Gene Xpert /Xpert ultra is the diagnostic test for drug resistant TB. Close to 90% of the participants knew that uncomplicated pulmonary TB is treated for 6 months, 43% knew the recommended regimen for uncomplicated TB and 46.4% knew that steroids are indicated in children with TB meningitis. There was low to moderate knowledge on indications for TPT in children, TPT regimens in Zambia and evaluation before start of TPT.

### Attitude about childhood TB

The median attitude score was 7 (IQR = 6.0–8.0) (Table 3). Overall, 48.1% had a good attitude. The majority of participants reported a proactive role in diagnosis of children with presumptive TB. Almost all participants reported compassion and a desire to help children with TB (97.1%) and 95.4% expressed willingness to be more involved in TB activities. About 5% of participants would either be angry or scared if they were assigned to work at the TB corner. The most frequent concerns about TPT were pill burden (28.7.3%) and potential side effects (46%). Two-thirds of the healthcare workers believed that benefits of TPT outweigh the risks.

**Table 1. Characteristics of the study participants.**

| Characteristics | Frequency *(n)* | Percentage *(%)* |
|---|---|---|
| **Age Group (years)** | | |
| Under 30 | 151 | 63.7 |
| 31–40 | 61 | 25.7 |
| 41–50 | 16 | 6.8 |
| Over 50 | 7 | 3.0 |
| Missing | 2 | 0.8 |
| **Sex** | | |
| Male | 65 | 27.4 |
| Female | 172 | 72.6 |
| **Profession** | | |
| Nurse | 167 | 70.5 |
| Clinical Officer | 53 | 22.4 |
| Medical Doctor | 17 | 7.2 |
| **Department** | | |
| OPD | 93 | 39.2 |
| ART | 27 | 11.4 |
| IPD | 74 | 31.2 |
| MCH | 38 | 16.0 |
| TB | 4 | 1.7 |
| Missing | 1 | 0.4 |
| **Duration of working at facility** | | |
| Less than 1 year | 109 | 46.0 |
| 1–5 years | 104 | 43.9 |
| 5 or more years | 24 | 10.1 |
| **Previous training on TB[3]** | | |
| No | 134 | 56.5 |
| Yes | 100 | 42.2 |
| Missing | 3 | 1.3 |
| **Focus of previous training** | | |
| Adult TB | 47 | 47.0 |
| Childhood TB | 35 | 35.0 |
| 3 Is | 9 | 9.0 |
| TB preventive therapy | 40 | 40.0 |
| Infection control | 29 | 29.3 |
| Case finding | 12 | 12.0 |
| TB/HIV management | 53 | 53.0 |
| **Duration since previous training** | | |
| Less than 6 months | 43 | 43.0 |
| 6 months to 2 years | 30 | 30.0 |
| More than 2 years | 27 | 27.0 |

Knowledge on childhood TB.

## Childhood TB-related practices

Then mean practice score was 5 points (SD = 1.9) (Table 4). Overall, 54.4% had good childhood TB practice. Gaps in regular educational practices were recorded, only 31.7% reported that their departments had provided health education on childhood TB diagnosis or

**Table 2. Knowledge of HCWs on childhood TB.**

| Knowledge questions | Responses | Frequency | Percentage |
|---|---|---|---|
| **TB epidemiology and transmission** | | | |
| **Zambia is a high burden TB country** | Yes | 211 | 89.0 |
| | No | 3 | 1.3 |
| | Not sure | 18 | 7.6 |
| | Missing | 5 | 2.1 |
| **TB transmission** | Coughing | 236 | 99.6 |
| | Sneezing | 167 | 70.5 |
| | Singing | 90 | 38.0 |
| | Laughing | 89 | 37.6 |
| | Skin contact | 15 | 6.3 |
| **Body parts affected by TB** | Lungs | 237 | 100 |
| | Larynx | 118 | 47.8 |
| | Heart | 100 | 42.4 |
| | Spine | 219 | 92.4 |
| | Meninges | 195 | 82.3 |
| | Lymph nodes | 168 | 71.2 |
| | Abdomen | 199 | 84.0 |
| | Pleura | 154 | 65.0 |
| **Extra-pulmonary TB is common in children** | Yes | 118 | 49.8 |
| | No | 96 | 40.5 |
| | Not sure | 10 | 4.2 |
| | Missing | 13 | 5.5 |
| **Risk factors for childhood TB infection** | Not being vaccinated with BCG | 211 | 90.2 |
| | HIV | 207 | 88.5 |
| | Malnutrition | 196 | 83.8 |
| | Being <5 years of age | 130 | 55.6 |
| | Living in Zambia | 70 | 30.0 |
| | Being a contact to a TB case | 203 | 86.8 |
| **Diagnosis of TB** | | | |
| **Symptoms of TB in children** | Cough | 226 | 96.7 |
| | Low appetite | 211 | 90.2 |
| | Tiredness/reduced playfulness | 198 | 84.6 |
| | Weight loss | 220 | 94.0 |
| | Fever | 204 | 87.2 |
| | Chest pain | 162 | 69.2 |
| | Shortness of breath | 176 | 75.2 |
| | Do not know | 0 | 0.0 |
| | Other | 7 | 3.0 |
| **Risk factors for childhood TB disease** | No BCG vaccination | 209 | 89.3 |
| | HIV | 213 | 91.1 |
| | Malnutrition | 196 | 83.8 |
| | Being less than 5 years old | 125 | 53.7 |
| | Living in Zambia | 45 | 19.3 |
| | Being a contact to a TB case | 200 | 85.5 |

(*Continued*)

**Table 2.** (*Continued*)

| Knowledge questions | Responses | Frequency | Percentage |
|---|---|---|---|
| **First line TB diagnostic test in Zambia** | Chest X-ray | 73 | 30.8 |
| | Gene Xpert/ Xpert ultra | 115 | 48.5 |
| | Smear microscopy | 43 | 18.1 |
| | Do not know | 2 | 0.8 |
| | Missing | 4 | 1.7 |
| **Number and type of samples needed for Gene Xpert testing** | One spot sample | 63 | 26.6 |
| | One morning sample | 78 | 32.9 |
| | Two samples: spot and morning | 67 | 28.3 |
| | Two spot samples | 4 | 1.7 |
| | Three samples: Spot, morning, spot | 14 | 5.9 |
| | Missing | 11 | 4.6 |
| **Diagnostic tests for diagnose drug resistant TB** | Chest x-ray | 77 | 32.6 |
| | Gene Xpert /Xpert ultra | 213 | 90.5 |
| | Smear microscopy | 106 | 44.9 |
| | Do not know | 24 | 10.2 |
| | Other | 1 | 0.4 |
| **Treatment of TB** | | | |
| **Duration of treatment of uncomplicated pulmonary TB in children** | 1 month | 5 | 2.1 |
| | 6 months | 210 | 88.6 |
| | 12 months | 9 | 3.8 |
| | Not sure | 11 | 4.6 |
| | Missing | 2 | 0.8 |
| **Treatment regimen for uncomplicated pulmonary TB in children** | 2HERZ/4HR | 102 | 43.0 |
| | 2HRZ/4HR | 96 | 40.5 |
| | 2HERZ/10HR | 12 | 5.1 |
| | 2HRZ/10HR | 10 | 4.2 |
| | Missing | 17 | 7.2 |
| **TB and HIV treatment can be started on the same day in children** | Yes | 62 | 26.2 |
| | No | 154 | 65.0 |
| | Not sure | 19 | 8.0 |
| | Missing | 2 | 0.8 |
| **Children with TB meningitis must always be given steroids** | Yes | 110 | 46.4 |
| | No | 58 | 24.5 |
| | Not sure | 62 | 26.2 |
| | Missing | 7 | 3.0 |
| **Prevention of TB** | | | |
| **Children eligible for TB preventive therapy (TPT)** | Children living with HIV (CLHIV) irrespective of age | 158 | 67.0 |
| | CLHIV > 1 year | 105 | 44.5 |
| | All children | 61 | 25.9 |
| | CLHIV <1 year in contact to a TB case | 127 | 53.8 |
| | < 5 contacts to bacteriologically confirmed TB cases | 129 | 54.7 |
| **Recommended TPT regimens in Zambia** | Ethambutol for 6 months | 19 | 8.1 |
| | 3H | 56 | 24.0 |
| | 6H | 153 | 64.8 |
| | 3RH | 37 | 15.7 |
| | Pyrazinamide for 3 months | 12 | 5.1 |
| | 3HP in children > 2 years | 19 | 8.1 |

(*Continued*)

**Table 2.** (Continued)

| Knowledge questions | Responses | Frequency | Percentage |
|---|---|---|---|
| **Children with symptoms of TB can be started on TB preventive therapy** | Yes | 78 | 32.9 |
| | No | 134 | 56.5 |
| | Not sure | 22 | 9.3 |
| | Missing | 3 | 1.3 |
| **Before starting TB preventive therapy, the following must be done** | Gene Xpert | 163 | 69.7 |
| | Chest x-ray | 153 | 65.1 |
| | Symptom screening | 190 | 81.2 |
| | Mantoux | 44 | 18.7 |
| **Overall knowledge level** | **Median Knowledge score** = 28 points (IQR 24.0–31.0) | | |
| | **Good knowledge** = 43.5% (103/237) **Poor knowledge** = 56.5% (134/237) | | |

Abbreviations: BCG- Bacillus Calmette–Guérin; HIV- Human Immunodeficiency virus; TB- Tuberculosis; 2HERZ/4HR- 2months of isoniazid, ethambutol, rifampicin and pyrazinamide followed by 4 months of isoniazid and rifampicin; 2HRZ/4HR- 2months of isoniazid, rifampicin and pyrazinamide followed by 4 months of isoniazid and rifampicin; 2HERZ/10HR- 2months of isoniazid, ethambutol, rifampicin and pyrazinamide followed by 10 months of isoniazid and rifampicin; 2HRZ/10HR-2months of isoniazid, rifampicin and pyrazinamide followed by 10 months of isoniazid and rifampicin; 3H- 3 months of isoniazid; 6H- 6 months of isoniazid; 3HR- 3 months of isoniazid and rifampicin; 3HP- weekly dose of isoniazid and rifapentine for 12 weeks.

prevention in the past week. Of the participants, 61.7% reported to interact with children exhibiting TB symptoms at least once a week. The most common practices when a child is identifying with presumptive TB is identified is to send them to the TB corner to submit sputum sample (74.4%).

In instances where children are not able to produce sputum, majority of HCWs (80.5%) reported collecting gastric aspirates.

## Association between characteristics of study participants and Knowledge, attitude, and practices in childhood TB

In the univariable analysis (Table 5), those working in IPD and MCH had lower odds of good knowledge about childhood TB: 0.40 (95% CI = 0.21–0.76, p = 0.005) and 0.36 (95% CI = 0.16–0.80, p = 0.013) respectively. On the other hand, clinical officers and previously undergoing training on TB had 2.82 higher odds (95% CI = 1.49–5.31, p = 0.001) and 1.95 higher odds (95% CI = 1.15–3.31, p = 0.013) respectively of having good knowledge. Regarding attitude, those working in MCH had 0.34 lower odds (95% CI = 0.15–0.78, p = 0.011) of good attitude while clinical officers and those with 1–5 years of working at the facility had higher odds of good attitude: 1.99 times (95% CI = 1.05–3.76, p = 0.034) and 2.58 times (95% CI = 1.48–4.49, p = 0.001) respectively. In terms of practice, participants trained on childhood TB more than 2 years ago had 0.25 times lower odds (95% CI =, 0.09–0.70, p = 0.008) of good practice compared to those trained less than 6 months ago. Other participant characteristics did not show statistically significant associations with the scores.

In the multivariate analysis (Table 6), clinical officers and individuals with 1–5 years work experience at the facility demonstrated higher odds, 2.61 (95% CI = 1.18–5.80, p = 0.018) and 3.09 (95% CI = 1.69–5.65, p = 0.001) respectively, of having good attitude. With regards to good practice, medical doctors had 0.17 lower odds (95% CI = 0.18–5.80, p = 0.018). However, the other participant characteristics didn't show a significant association with the scores.

**Table 3. Attitudes of HCWs towards childhood TB.**

| Childhood TB Attitude items | Responses | Frequency | Percentage |
|---|---|---|---|
| **What do you feel is your role in diagnosis children with TB?** | Refer children with presumptive TB to the TB corner | 203 | 86.4 |
| | Request children with presumptive TB to submit sputum | 172 | 73.2 |
| | Fast track the children so that they can see a clinician quickly | 148 | 63.0 |
| | Document these children in the presumptive TB register | 148 | 63.0 |
| | I don't know | 3 | 1.3 |
| **Which statement is closest to your feeling about children with TB disease?** | I feel compassion and desire to help | 230 | 97.1 |
| | I feel compassion but I tend to stay away from these people | 1 | 0.4 |
| | It's their problem, and I cannot get TB | 1 | 0.4 |
| | I fear them because they may infect me | 1 | 0.4 |
| | I have no particular feeling | 1 | 0.4 |
| | Other | 1 | 0.4 |
| | Missing | 3 | 1.3 |
| **Would you like to be more involved in TB activities?** | Yes | 226 | 95.4 |
| | No | 3 | 1.3 |
| | Not sure | 6 | 2.5 |
| | Missing | 2 | 0.8 |
| **What would be your reaction if you were asked to work at the TB corner?** | I would refuse | 2 | 0.8 |
| | I don't mind | 138 | 58.2 |
| | Scared | 9 | 3.8 |
| | Angry | 2 | 0.8 |
| | Happy | 76 | 32.1 |
| | Other | 7 | 3.0 |
| | Missing | 3 | 1.3 |
| **What is your biggest fear/concern about TPT?** | Pill burden | 68 | 28.7 |
| | TB preventive therapy promotes drug resistant TB | 42 | 17.7 |
| | It does not have much benefit in a high burden setting | 3 | 1.3 |
| | Side effects | 109 | 46.0 |
| | Other | 5 | 2.1 |
| | Missing | 10 | 4.2 |
| **Do you think the benefits of TB preventive therapy outweigh the risks?** | Yes | 174 | 73.4 |
| | No | 33 | 13.9 |
| | Not sure | 25 | 10.6 |
| | Missing | 5 | 2.1 |
| **Overall attitudes levels** | **Median attitude score** = 7 (IQR = 6.0–8.0) | | |
| | **Good attitude** = 48.1 (114/237) **Poor attitude** = 51.9 (123/237) | | |

## Discussion

In this cross-sectional study, we found that a high proportion of HCWs with poor knowledge, attitudes and practices towards childhood TB. The knowledge gaps were most pronounced in the domains TB diagnosis and treatment. Being a clinical officer and work experience of 1–5 years at the facility were associated with good attitude regarding childhood TB while being a medical doctor was associated with poor practice regarding childhood TB.

Our study findings were aligned to the performance on childhood TB case detection in the study sites which was only 4–6% of the total notifications at the time, a performance significantly lower than the target of 10–15% of total notifications in high TB burden settings [1,16].

**Table 4. Practices of HCWs on childhood TB.**

| Childhood TB practices questions | Responses | Frequency | Percentage |
|---|---|---|---|
| Has your department provided any health education on childhood TB diagnosis or prevention in the past 1 week? | Yes | 75 | 31.7 |
| | No | 145 | 61.2 |
| | Not sure | 15 | 6.3 |
| | Missing | 2 | 0.8 |
| How often do you interact with children with symptoms suggestive of TB? | At least once week | 142 | 59.9 |
| | Once a month | 48 | 20.3 |
| | Never | 40 | 16.9 |
| | Missing | 7 | 3.0 |
| What do you do when you identify a child with symptoms of TB? | Ask the patient to go to TB corner to submit a sputum sample | 174 | 74.4 |
| | Request the patient to submit a sputum sample | 131 | 55.7 |
| | Fast track the child | 132 | 56.7 |
| | Provide education on cough etiquette | 138 | 59.0 |
| | Document the child in the presumptive TB register | 100 | 42.7 |
| | Nothing | 2 | 0.9 |
| What do you do for children not able to produce sputum? | Prescribe antibiotics for 1 week and then ask them to return to the health facility for review | 18 | 7.8 |
| | Collect gastric aspirates | 186 | 80.5 |
| | Use chest x-ray to make a diagnosis | 121 | 52.4 |
| | Give mother sputum bottle to continue trying to get sputum from the child | 36 | 15.7 |
| | Request for LAM | 52 | 22.5 |
| | Other | 5 | 2.2 |
| | N/A | 1 | 0.4 |
| **Overall practice** | **Mean practice score** = 5 points (SD = 1.9) | | |
| | **Good practice** = 54.4 (129/237) **Poor practice** = 45.6 (108/237) | | |

Overall, our findings are similar to those from other childhood TB KAP studies [9,10,17]. However, it's worth noting that the multi-country study conducted in Cambodia, Cameroon, Cote d'Ivoire, Sierra Leone and Uganda found more favourable attitudes than we did [10] and the percentage of participants with good knowledge, attitudes and practices was slightly higher in the studies conducted in Cameroon and Saudi Arabia [9,17]. These differences could be due to variations in settings as well as in the questionnaire. Additionally, our findings are similar to those from TB KAP surveys that were not specifically focused on childhood TB [18–20].

The very low KAP scores in our study could be attributed to the fact that a majority of healthcare workers had no previous training in TB, a gap most pronounced in childhood TB. This is secondary to the fact that TB programs in Zambia have traditionally focused on training those working at the TB department. However, this approach inadvertently creates a gap, as majority of the presumptive TB patients, including childhood presumptive TB patients, are identified outside the TB department, most especially at OPD and ART [21–23]. Much as the gap on treatment is significant, it is low impact is somewhat mitigated by the existing procedural norms. Specifically, while patients identified with TB in the IPD can commence treatment within the IPD itself, those diagnosed in other departments are mandated to initiate their TB treatment at the chest clinic.

Although we didn't find any participant characteristics that were associated with good knowledge towards childhood TB, evidence from other settings suggests that previous training is associated with increased knowledge in childhood TB [9,24] and duration of work

**Table 5. Univariable logistic regression analysis of characteristics with Knowledge, attitude, and practices in childhood TB.**

| Characteristics | Knowledge score | | Attitude score | | Practice score | |
|---|---|---|---|---|---|---|
| | Unadjusted odds ratio (95% CI) | *p*-value | Unadjusted odds ratio (95% CI) | *p*-value | Unadjusted odds ratio (95% CI) | *p*-value |
| **Age group** (years) | | | | | | |
| Under 30 | Ref. | | Ref. | | Ref. | |
| 31–40 | 1.01 (0.55–1.84) | 0.975 | 1.12 (0.62–2.03) | 0.713 | 1.28 (0.70–2.33) | 0.425 |
| 41–50 | 1.06 (0.37–2.99) | 0.916 | 0.46 (0.15–1.39) | 0.169 | 0.69 (0.24–1.95) | 0.484 |
| Over 50 | 3.40(0.64–18.08) | 0.151 | 0.41 (0.08–2.15) | 0.289 | 5.33 (0.63–45.30) | 0.126 |
| **Sex** | | | | | | |
| Male | Ref. | | Ref. | | Ref. | |
| Female | 0.79 (0.44–1.40) | 0.413 | 0.81 (0.46–1.44) | 0.472 | 0.69 (0.38–1.23) | 0.207 |
| **Profession** | | | | | | |
| Nurse | Ref. | | Ref. | | Ref. | |
| Clinical Officer | 2.83 (1.49–5.37) | <0.002 | 1.99 (1.05–3.76) | <0.034 | 1.32 (0.70–2.49) | 0.395 |
| Medical Doctor | 2.52 (0.91–6.97) | 0.074 | 1.12 (0.41–3.01) | 0.845 | 0.45 (0.16–1.27) | 0.132 |
| **Department** | | | | | | |
| OPD | Ref. | | Ref. | | Ref. | |
| ART | 0.78 (0.33–1.84) | 0.571 | 1.56 (0.65–3.76) | 0.324 | 1.63 (0.67–3.93) | 0.279 |
| IPD | 0.40 (0.21–0.76) | <0.005 | 0.82 (0.45–1.52) | 0.532 | 1.40 (0.76–2.61) | 0.282 |
| MCH | 0.36 (0.16–0.80) | <0.013 | 0.34 (0.15–0.78) | <0.011 | 0.81 (0.38–175) | 0.598 |
| TB | Omitted | Omitted | 2.75 (0.28–27.42) | 0.389 | Omitted | Omitted |
| **Duration working at facility** | | | | | | |
| Less than 1 year | Ref. | | Ref. | | Ref. | |
| 1–5 years | 1.23 (0.71–2.12) | 0.461 | 2.58 (1.48–4.49) | <0.001 | 1.19 (0.69–2.05) | 0.522 |
| 5 or more years | 1.02 (0.42–2.5175) | 0.961 | 0.97 (0.39–2.41) | 0.941 | 1.28 (0.52–3.12) | 0.595 |
| **Previous training on TB** | | | | | | |
| No | Ref. | | Ref. | | Ref. | |
| Yes | 1.95 (1.15–3.31) | <0.013 | 1.28 (0.76–2.16) | 0.351 | 1.12 (0.66–1.89) | 0.671 |
| **Duration since previous training** | | | | | | |
| Less than 6 months | Ref. | | Ref. | | Ref. | |
| 6 months to 2 years | 0.55(0.21–1.41) | 0.215 | 0.87 (0.34–2.21) | 0.769 | 0.57(0.21–1.50) | 0.252 |
| More than 2 years | 09.(0.34–2.38) | 0.832 | 0.94 (0.36–2.46) | 0.894 | 0.25 (0.09–0.70) | <0.008 |

*P value <0.005.

experience is associated with a good attitude [9]. Additional evidence from the same setting [13] and other settings [25–27] indicates that capacity development of healthcare workers leads to improved childhood TB case detection. The improvement in case detection is likely driven by a prior enhancement of KAP among healthcare professionals. The positive association between being a clinical officer and having 1–5 years' experience with a good attitude towards childhood tuberculosis suggests that frontline healthcare providers with moderate levels of experience exhibit more favorable attitudes, possibly due to a combination of practical experience, ongoing learning, and familiarity with the local context. Contrastingly, the association between being a medical doctor and poor practice regarding childhood tuberculosis raises important questions. The potential factors contributing to this finding include heavy workload requiring them to focus on managing other disease conditions and potential gaps in on-the-job training specifically related to childhood tuberculosis.

The strengths of our study include a high response rate and inclusion of departments regularly involved in TB screening and diagnosis which provides true insights into the childhood

**Table 6. Multivariable logistic regression analysis of characteristics with Knowledge, attitude, and practices in childhood TB.**

| Characteristics | Knowledge score | | Attitude score | | Practice score | |
|---|---|---|---|---|---|---|
| | Adjusted odds ratio (95% CI) | *p*-value | Unadjusted odds ratio (95% CI) | *p*-value | Unadjusted odds ratio (95% CI) | *p*-value |
| **Age group** | | | | | | |
| Under 30 | - | - | Ref. | | Ref. | |
| 31–40 | - | - | 1.28 (0.63–2.60) | 0.490 | 2.69 (0.87–8.31) | 0.085 |
| 41–50 | - | - | 0.31 (0.08–1.26) | 0.103 | 1.05 (0.20–5.41) | 0.956 |
| Over 50 | - | - | 0.33 (0.04–2.65) | 0.299 | Omitted | Omitted |
| **Sex** | | | | | | |
| Male | - | - | - | - | Ref. | |
| Female | - | - | - | - | 0.60 (0.22–1.61) | 0.311 |
| **Profession** | | | | | | |
| Nurse | Ref. | | Ref. | | Ref. | |
| Clinical Officer | 1.94 (0.93–4.07) | 0.078 | 2.61 (1.18–5.80) | <0.018 | 0.83 (0.30–2.29) | 0.717 |
| Medical Doctor | 2.25 (0.78–4.07) | 0.134 | 0.96 (0.32–2.85) | 0.940 | 0.17 (0.03–0.97 | <0.046 |
| **Department** | | | | | | |
| Outpatient department | Ref. | | Ref. | | - | - |
| ART | 0.82 (0.33–2.05) | 0.675 | 1.75 (0.66–4.66) | 0.259 | - | - |
| In Patient Department | 0.52 (0.26–1.07) | 0.074 | 1.09 (0.54–2.22) | 0.812 | - | - |
| MCH | 0.51 (0.21–1.25) | 0.139 | 0.48 (0.19–1.26) | 0.136 | - | |
| TB | Omitted | Omitted | 9.97 (0.81–123.03) | 0.073 | - | - |
| **Duration working at facility** | | | | | | |
| Less than 1 year | - | - | Ref. | | - | - |
| 1–5 years | - | - | 3.09 (1.69–5.65) | <0.001 | - | - |
| 5 or more years | - | - | 1.50 (0.45–5.03) | 0.513 | - | - |
| **Previous training on TB** | | | | | | |
| No | Ref. | | - | - | - | - |
| Yes | 1.46 (0.82–2.60) | 0.203 | - | - | - | - |
| **Duration since previous training** | | | | | | |
| Less than 6 months | - | - | - | - | Ref. | |
| 6 months to 2 years | - | - | - | - | 0.79 (0.27–2.31) | 0.669 |
| More than 2 years | - | - | - | - | 0.18 (0.06–0.56) | <0.003 |

*$P$ value <0.005.

TB KAP among frontline healthcare professionals. Additionally, this is the first KAP study on childhood TB in Zambia. However, out study has important limitations: we didn't test the validity and reliability of the questionnaire and the findings from this study cannot be generalized to Zambia.

In conclusion, this study found high levels of poor KAP towards childhood TB at 2 primary health care facilities servicing high burden TB communities. The identified knowledge gaps, attitudes, and practices provide a foundation for strategic interventions. To address these gaps, it is important that the ministry of health (MOH) collaborates with key stakeholders, including non-governmental organizations (NGOs) and the respective study facilities.

## Supporting information

**S1 Dataset.**
(XLSX)

**S1 File.**
(DOCX)

## Author Contributions

**Conceptualization:** Mary Kagujje, Sarah Nyangu, Monde Muyoyeta.

**Data curation:** Paul Chabala Kaumba, Mary Kagujje, Minyoi Mubita Maimbolwa, Monde Muyoyeta.

**Formal analysis:** Paul Chabala Kaumba, Daniel Siameka, Mary Kagujje, Monde Muyoyeta.

**Funding acquisition:** Monde Muyoyeta.

**Investigation:** Paul Chabala Kaumba, Mary Kagujje, Sarah Nyangu, Brian Shuma, Monde Muyoyeta.

**Methodology:** Paul Chabala Kaumba, Mary Kagujje.

**Project administration:** Mary Kagujje, Chalilwe Chungu, Sarah Nyangu, Nsala Sanjase, Brian Shuma, Lophina Chilukutu, Monde Muyoyeta.

**Resources:** Monde Muyoyeta.

**Software:** Paul Chabala Kaumba, Minyoi Mubita Maimbolwa.

**Supervision:** Mary Kagujje, Chalilwe Chungu, Monde Muyoyeta.

**Validation:** Paul Chabala Kaumba, Mary Kagujje, Monde Muyoyeta.

**Visualization:** Paul Chabala Kaumba.

**Writing – original draft:** Paul Chabala Kaumba, Sarah Nyangu.

**Writing – review & editing:** Paul Chabala Kaumba, Mary Kagujje, Chalilwe Chungu, Sarah Nyangu, Nsala Sanjase, Minyoi Mubita Maimbolwa, Brian Shuma, Lophina Chilukutu, Monde Muyoyeta.

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
