## [Decision Letter · Decision Letter 0]

29 Aug 2023

PONE-D-23-18137Health Care Workers Knowledge, Attitudes, and Practice Towards Childhood Tuberculosis in Primary Health Facilities in Lusaka, Zambia.PLOS ONE

Dear Dr. Kaumba,

Thank you for submitting your manuscript to PLOS ONE. After careful consideration, we feel that it has merit but does not fully meet PLOS ONE’s publication criteria as it currently stands. Therefore, we invite you to submit a revised version of the manuscript that addresses the pointsicate which changes you require for acceptance versus which changes you recommend.Please submit your revised manuscript by Oct 13 2023 11:59PM. If you will need more time than this to complete your revisions, please reply to this message or contact the journal office at plosone@plos.org. Please include the following items when submitting your revised manuscript:A rebuttal letter that responds to each point raised by the academic editor and reviewer(s). You should upload this letter as a separate file labeled 'Response to Reviewers'.A marked-up copy of your manuscript that highlights changes made to the original version. You should upload this as a separate file labeled 'Revised Manuscript with Track Changes'.An unmarked version of your revised paper without tracked changes. You should upload this as a separate file labeled 'Manuscript'.

We look forward to receiving your revised manuscript.

Kind regards,

Khin Thet Wai, MBBS, MPH, MA

Academic Editor

PLOS ONE

“None Declared”

Additional Editor Comments:

- English language editing is deemed necessary.

- Authors need to improve the overall scientific writeup by following STROBE Guidelines.

- The methodology part requires an extensive revision.

Reviewers' comments:

Reviewer's Responses to Questions

**Comments to the Author**

1. Is the manuscript technically sound, and do the data support the conclusions?

Reviewer #1: Yes

Reviewer #2: No

2. Has the statistical analysis been performed appropriately and rigorously? 

Reviewer #1: Yes

Reviewer #2: No

3. Have the authors made all data underlying the findings in their manuscript fully available?

Reviewer #1: Yes

Reviewer #2: Yes

4. Is the manuscript presented in an intelligible fashion and written in standard English?

Reviewer #1: Yes

Reviewer #2: No

5. Review Comments to the Author

Reviewer #1: A. Minor comments-

1. Line 103- Knowledge of participant- Is this on TB or only on childhood TB? Please specify based on your KAP questionnaire.

2. Line 111- Please write full form of CIDRZ if you have not mentioned prior.

3. Please check line 201 and 202. The sentence is not aligned properly.

4. Line 265-66- Among the variables, the significant association was found between KAP scores and Sex, Department, Trained- Please re-write this sentence to make it clearer.

5. Line 291-92- "Staff from OPD, ART, and MCH had less understanding of TB infection control than those who did"- This sentence is not clear. What do you mean by "who did" in this sentence.

6. Better to present all responses on KAP questionnaire in the form of table in supplementary data section.

B. Major comments-

METHODS-

1. Its better if you also mention the number of questions (out of 37) on Knowledge, Attitudes and Practices were there in KAP questionnaire.

2. In statistical analysis part, you have only explained how you calculated Knowledge score but you have not mentioned about the scores for Attitudes and Practices. Please explain.

3. Which data analysis software did you use for the analyisis of KAP data? Please do mention.

4. You have mentioned "Passed" and "Failed" in Result section but what is your criteria for this classification. You should mention this in methods section.

RESULTS-

5. I can see questions on Knowledge and practices in table 2 and 3 respectively but I cant see questions on Attitudes. Can you also present some questions on Attitudes.

6. In table 4 there is high non-response (more than 60%) rate for the question on TB infection control. What might be the reason for this? Was there high non-response rate for other questions as well?

7. Heading of table 6 is not in line with data you presented in table as you have also presented Attitudes and Practices scores in table.

DISCUSSION-

8. Can you please include some references in discussion section. Some paragraphs in discussion sections are without any references. Its better to have references to support your findings in discussion section.

Reviewer #2: The current study assesses the knowledge attitude and practices of HCWs regarding childhood TB. The topic is relevant and of global public health importance. However, I have the following comments and suggestions

1. General comments:

• Change health care to one word, healthcare on the study title and entire manuscript.

• Too many grammatical errors in the manuscript with poor punctuation: Full stops are included before the end of the reference in the majority of the sentences in the introduction section. Capital letter in the course of a sentence, lines 119, 218, 220, 266, 284, 351

• The authors should improve the quality of the manuscript, especially on the methodology by reading recently published papers on childhood TB KAP studies between 2021 and 202

https://pubmed.ncbi.nlm.nih.gov/36569997/

https://pubmed.ncbi.nlm.nih.gov/35361143/

https://pubmed.ncbi.nlm.nih.gov/35197164/

• The objectives of the study are not clear and the results are not coherent, not objectively presented within the text.

2. Abstract

• Abstract too long, more than 350 words, kindly review.

• Line 24 and 24: As per the WHO 2022 annual report, page 42, Zambia joined the list of 30 high MDR/RR-TB burden countries in 2022. So the sentence should read as Zambia is among the list of 30 high-burden countries for TB, HIV-associated TB, and MDR/RR-TB with a significant…

• Line 31: Add childhood before TB

• Line 32: Please indicate how the HCWs and or health facilities were sampled. Random sampling of the population is mentioned only under the strengths of the study

• Line 33: The phrase “the questionnaire data collected was later transcribed to an electronic system called DHIS 2” is not clear and confusing. Please kindly state how the data was analysis including the level of significance

• Line 34 and 45: The results are too long and do not present any finding on the study objective which is to understand the factors influencing the knowledge, 28 attitudes, and practices of health care workers (HCWs) regarding childhood TB in Zambia. Please kindly review. In addition, the conclusion is too long and does not correlate with the study findings.

• Line 56: Please remove survey as a keyword, not mentioned in the abstract or title of the manuscript.

Introduction

Line 61: The sentence “Majority (about 96%) of children who died from TB were not on treatment 62 (2)” is data from 2017 and does not correlate with the previous statement on globally…. Also, the sentence needs to be updated with data from the WHO 2022 TB annual report.

Line 63: The phrase should be made to children… is not clear and should be changed to should be among children

Line: 69: Pediatric TB diagnosis can be difficult. (4-6) should be corrected by including a full stop instead at the end of the sentence

Line 78, 80: Full stop at the end of the reference

Line 83: The objective mentioned is different from that at the level of the abstract. Please could you clarify on this?

Methods

• The methodology has a lot of missing information. Sampling of facility and HCWs, sample size, variables of interest, and not coherent

• Measurements: Your methodology does not explain how knowledge level was classified as bad, good, and unknown

• I recommend the authors should follow the STROBE guidelines

Line 87: What were the eligibility criteria for the HCWs to participate in the study? How were these HCWs sampled? The authors mentioned that registered healthcare workers (nurses, clinical officers, and medical officers) working at the health facility participated in the study. How sure are you that HCWs working in service units such as surgery, dentistry, and diabetic units for more than two years could provide correct information on childhood TB knowledge attitude, and practice?. I do not concur with your selection criteria.

Line 98: Please reference the different sources used to adopt the questionnaire. How did you ensure the validity of the questionnaire? Was it pre-tested? What about the different internal constructs of the questionnaire? Was Cronbach's alpha coefficient used to measure the internal consistency, or reliability, of the survey items?

Results:

Line 161: The different types of departments should be written in full to ease understanding or you indicated the full meaning at the end of the table

All the tables should be formatted in the standard way with three horizontal lines. See the example below for table 1

Table 1: Sociodemographic characteristics of the study population

Variable Frequency (N) Percentage (%)

Age Group

Under 30

41-50

Over 50

Unknown

Sex

Male 66 27.7

Female 172 72.3

Department

Outpatient Department

XXX

XXX

XXX

XXX

Duration employed

Training

The variable “How long ago were you trained?” should be changed to training duration and the five modalities should be reduced to at most three (eg less than six months, Six months to two years, more than two years) in order to ease understanding and appreciate the data

Line 163: Include a heading on HCW knowledge

Line 202: sentence not clear please correct

210: I don’t think we say close association in scientific writing.

Line 214-216: The sentence “The characteristics and scores of healthcare workers in each category of the Knowledge, Attitudes, and Practices (KAP) survey are presented in Table 6: HCWs characteristics and 216 knowledge score below and in Table 5:” is not clear, please correct

Discussion

The discussion section is poorly written as references to similar studies are lacking. There is evidence of a poor literature review done on the paper.

References

The number of references (20) for this manuscript shows much literature was not done whereas many studies have been published on this topic between 21 and 22.

Line 374, 378, and 425: References are incomplete, kindly review

Line 67: Add a full stop at the end of the sentence

6. PLOS authors have the option to publish the peer review history of their article (what does this mean?). If published, this will include your full peer review and any attached files.

Reviewer #1: No

Reviewer #2: No

---

## [Author Response · Author response to Decision Letter 0]

5 Feb 2024

Dear Editor,

First and foremost, we would like to thank the you and the reviewers for taking time to review the submitted manuscript. We value the insightful comments provided, as addressing them has strengthened the quality of the manuscript. Our point-by-point responses to the comments are detailed in bold below.

EDITOR’S COMMENTS

and

RESPONSE: The manuscript has been formatted to follow this guideline.

“None Declared”

RESPONSE: Thanks. This will be done when resubmitting the manuscript.

RESPONSE: The data will be made available as part of this publication.

RESPONSE: This has been done, thank you. 

RESPONSE: It was in the methods section, thank you

REVIEWER 1

1. Line 103- Knowledge of participant- Is this on TB or only on childhood TB? Please specify based on your KAP questionnaire

RESPONSE: Thank you for the comment. This section has been restructured with that particular text removed. However, we have clarified that this was in relation to childhood TB.

2. Line 111- Please write full form of CIDRZ if you have not mentioned prior.

RESPONSE: The sentence has been revised, with the removal of the term CIDRZ.

3. Please check line 201 and 202. The sentence is not aligned properly.

RESPONSE: The sentence has been removed from the updated results section.

4. Line 265-66- Among the variables, the significant association was found between KAP scores and Sex, Department, Trained- Please re-write this sentence to make it clearer.

RESPONSE: This has been resolved with the updated results section.

5. Line 291-92- "Staff from OPD, ART, and MCH had less understanding of TB infection control than those who did"- This sentence is not clear. What do you mean by "who did" in this sentence.

RESPONSE: This has been resolved with the removal of the sentence in the updated results section.

6. Better to present all responses on KAP questionnaire in the form of table in supplementary data section.

RESPONSE: Thanks for the guide, all responses have been presented in the form of a table. 

7. Its better if you also mention the number of questions (out of 37) on Knowledge, Attitudes and Practices were there in KAP questionnaire.

RESPONSE: This has been done, please see line 109-111.

8. In statistical analysis part, you have only explained how you calculated Knowledge score but you have not mentioned about the scores for Attitudes and Practices. Please explain.

RESPONSE: This has been addressed. Please see lines 124-131. 

9. Which data analysis software did you use for the analysis of KAP data? Please do mention.

RESPONSE: STATA Statistical Software (Stata Corporation Version 14. College Station, Texas 77845, USA) was used. This has been included on line 120-121.

10. You have mentioned "Passed" and "Failed" in Result section but what is your criteria for this classification. You should mention this in methods section.

RESPONSE: The criteria has been added to the methods section. Please see line 128-131.

11. I can see questions on Knowledge and practices in table 2 and 3 respectively but I cant see questions on Attitudes. Can you also present some questions on Attitudes.

RESPONSE: This has been addressed in the revised result section. Please see line 188.

12. In table 4 there is high non-response (more than 60%) rate for the question on TB infection control. What might be the reason for this? Was there high non-response rate for other questions as well?

RESPONSE: We did not have a high non-response rate. In fact, overall, missing results are less than 5%. We have changed the presentation of the results to improve the clarity. Also, we have attached the questionnaire to provide insights into the nature of the questions. 

13. Heading of table 6 is not in line with data you presented in table as you have also presented Attitudes and Practices scores in table.

RESPONSE: This has been addressed.

Discussion

14. 8. Can you please include some references in discussion section. Some paragraphs in discussion sections are without any references. Its better to have references to support your findings in discussion section.

RESPONSE: This has been addressed, thanks

REVIEWER 2

1. Change health care to one word, healthcare on the study title and entire manuscript.

RESPONSE: Thank you. This change has been made on the title and the rest of the document.

2. Too many grammatical errors in the manuscript with poor punctuation: Full stops are included before the end of the reference in the majority of the sentences in the introduction section. Capital letter in the course of a sentence, lines 119, 218, 220, 266, 284, 351

RESPONSE: These have been addressed. Some sentences have been reworded altogether to improve the grammar.

3. The authors should improve the quality of the manuscript, especially on the methodology by reading recently published papers on childhood TB KAP studies between 2021 and 202

https://pubmed.ncbi.nlm.nih.gov/36569997/

https://pubmed.ncbi.nlm.nih.gov/35361143/

https://pubmed.ncbi.nlm.nih.gov/35197164/

RESPONSE: We appreciate these resources; they have been useful in improving our methodology and results sections. 

4. The objectives of the study are not clear and the results are not coherent, not objectively presented within the text.

RESPONSE: We have revised the objectives to make them clear. The sentence on objectives currently reads as: “We undertook a study to assess healthcare workers’ (HCWs) knowledge, attitudes, and practices towards childhood TB and the factors associated with good knowledge, attitudes, and practices towards childhood TB.” Please see line 91-94. We have also revised our results to align more closely with these objectives. Please review the results section for the relevant updates. 

5. Abstract too long, more than 350 words, kindly review.

RESPONSE: The abstract has been revised to less than 350 words.

6. Line 24 and 24: As per the WHO 2022 annual report, page 42, Zambia joined the list of 30 high MDR/RR-TB burden countries in 2022. So the sentence should read as Zambia is among the list of 30 high-burden countries for TB, HIV-associated TB, and MDR/RR-TB with a significant…

RESPONSE: Thank you, we have revised the sentence as recommended. Please see line 24- 26.

7. Line 31: Add childhood before TB.

RESPONSE: Thank you, the sentence to which this comment corresponds has been removed.

8. Line 32: Please indicate how the HCWs and or health facilities were sampled. Random sampling of the population is mentioned only under the strengths of the study.

RESPONSE: Thank you. We have added that healthcare workers were selected through stratified random sampling. Please see line 32-33 (abstract section) as well as line 95 (methodology section).

9. Line 33: The phrase “the questionnaire data collected was later transcribed to an electronic system called DHIS 2” is not clear and confusing. Please kindly state how the data was analysis including the level of significance.

RESPONSE: Thank you, the sentence has been removed. Please see lines 31-37 on how data was analysed including the level of significance.

10. Line 34 and 45: The results are too long and do not present any finding on the study objective which is to understand the factors influencing the knowledge, 28 attitudes, and practices of health care workers (HCWs) regarding childhood TB in Zambia. Please kindly review. In addition, the conclusion is too long and does not correlate with the study findings.

RESPONSE: This is well noted. We have clarified the objective of the study in lines 27-29 and have restructured the results and conclusion to better align them to the study objectives. 

11. Line 56: Please remove survey as a keyword, not mentioned in the abstract or title of the manuscript.

RESPONSE: This has been done.

12. Line 61: The sentence “Majority (about 96%) of children who died from TB were not on treatment 62 (2)” is data from 2017 and does not correlate with the previous statement on globally…. Also, the sentence needs to be updated with data from the WHO 2022 TB annual report.

RESPONSE: We have restructured this paragraph and used the most recent TB data. The revised paragraph reads, “Globally, an estimated 10.6 million people developed tuberculosis (TB) in 2022 of which about 12% were children. Of these children, only 46% were diagnosed and started on treatment. In Zambia, one of the 30 high-burden countries for Tuberculosis (TB), Human Immunodeficiency Virus (HIV)-associated TB, and multi-drug resistant/rifampicin resistant TB, only 68% of the estimated children with TB were diagnosed and started on treatment in 2022. Between 2020 and 2021, this proportion did not exceed 50% in either year.” Please see lines 62-67.

13. Line 63: The phrase should be made to children… is not clear and should be changed to should be among children

RESPONSE: This is noted. The sentence has since been removed.

14. Line: 69: Pediatric TB diagnosis can be difficult. (4-6) should be corrected by including a full stop instead at the end of the sentence

RESPONSE: This was done. In addition, the correction was done to all other parts of the document that had this error.

15. Line 78, 80: Full stop at the end of the reference.

RESPONSE: Like the issue on number 14, this has addressed.

16. Line 83: The objective mentioned is different from that at the level of the abstract. Please could you clarify on this?

RESPONSE: This has addressed, the objectives have been aligned.

17. The methodology has a lot of missing information. Sampling of facility and HCWs, sample size, variables of interest, and not coherent

RESPONSE: This has addressed. More information has been added to the methodology section according to the STROBE guidelines and the requirements of the study.

18. Measurements: Your methodology does not explain how knowledge level was classified as bad, good, and unknown 

RESPONSE: We have revised our methodology to address this concern. First, we added a sub-section on validation of correct responses. See line 111-113. We then explained how knowledge level was classified as good or poor, see line 121-131. 

19. I recommend the authors should follow the STROBE guidelines.

RESPONSE: We appreciate your suggestion to adhere to the strobe guidelines. Following your recommendation, the necessary actions have been taken. Thank you.

20. Line 87: What were the eligibility criteria for the HCWs to participate in the study? How were these HCWs sampled? The authors mentioned that registered healthcare workers (nurses, clinical officers, and medical officers) working at the health facility participated in the study. How sure are you that HCWs working in service units such as surgery, dentistry, and diabetic units for more than two years could provide correct information on childhood TB knowledge attitude, and practice?. I do not concur with your selection criteria.

RESPONSE: We have clarified this- The selection criteria have been expanded on in the methodology. Basically, this involved all HCWs at departments that routinely provide TB screening and/or treatment of TB. Please see line 96-99.

21. Line 98: Please reference the different sources used to adopt the questionnaire. How did you ensure the validity of the questionnaire? Was it pre-tested? What about the different internal constructs of the questionnaire? Was Cronbach's alpha coefficient used to measure the internal consistency, or reliability, of the survey items?

RESPONSE: The questions were adapted primarily from the national pre-training test on childhood TB and the World Health Organisation (WHO) guide to developing KAP surveys. This has been added to line 108-109. We did not check the validated and reliability of the questionnaire. This has been included as a limitation. Please check line 276-278.

22. All the tables should be formatted in the standard way with three horizontal lines. See the example below for table 1.

RESPONSE: The tables have been edited according to the guidelines provided.

23. The variable “How long ago were you trained?” should be changed to training duration and the five modalities should be reduced to at most three (eg less than six months, Six months to two years, more than two years) in order to ease understanding and appreciate the data

RESPONSE: We have reworded “How long ago were you trained? to “Duration since previous training”. We have also reduced the responses to three. We have also applied this change to duration of working at the facility. Please see table 1 on line 154. 

24. Line 163: Include a heading on HCW knowledge.

RESPONSE: This has been done.

25. Line 202: sentence not clear please correct.

RESPONSE: The sentence was revised to make it clearer.

26. 210: I don’t think we say close association in scientific writing.

RESPONSE: The results section was revised as appropriate.

27. Line 214-216: The sentence “The characteristics and scores of healthcare workers in each category of the Knowledge, Attitudes, and Practices (KAP) survey are presented in Table 6: HCWs characteristics and 216 knowledge score below and in Table 5:” is not clear, please correct.

RESPONSE: This entire results section has been changed, with the removal of the sentence.

28. The discussion section is poorly written as references to similar studies are lacking. There is evidence of a poor literature review done on the paper.

RESPONSE: The discussion section was revised.

29. The number of references (20) for this manuscript shows much literature was not done whereas many studies have been published on this topic between 21 and 22.

RESPONSE: We have included done a more thorough literature review and have more references.

30. Line 374, 378, and 425: References are incomplete, kindly review.

RESPONSE: This was resolved.

31. Line 67: Add a full stop at the end of the sentence

RESPONSE: This was resolved.

Yours sincerely,

Paul Chabala Kaumba, Ph.D.

---

## [Decision Letter · Decision Letter 1]

23 Feb 2024

Knowledge, attitudes, and practices towards childhood tuberculosis among healthcare workers at two primary health facilities in Lusaka, Zambia

PONE-D-23-18137R1

Dear Dr. Kaumba,

We’re pleased to inform you that your manuscript has been judged scientifically suitable for publication and will be formally accepted for publication once it meets all outstanding technical requirements.

Kind regards,

Khin Thet Wai, MBBS, MPH, MA

Academic Editor

PLOS ONE

Additional Editor Comments (optional):

All comments are adequately addressed.

Reviewers' comments:

Reviewer's Responses to Questions

**Comments to the Author**

1. If the authors have adequately addressed your comments raised in a previous round of review and you feel that this manuscript is now acceptable for publication, you may indicate that here to bypass the “Comments to the Author” section, enter your conflict of interest statement in the “Confidential to Editor” section, and submit your "Accept" recommendation.

Reviewer #1: All comments have been addressed

Reviewer #2: All comments have been addressed

2. Is the manuscript technically sound, and do the data support the conclusions?

Reviewer #1: Yes

Reviewer #2: Yes

3. Has the statistical analysis been performed appropriately and rigorously? 

Reviewer #1: Yes

Reviewer #2: Yes

4. Have the authors made all data underlying the findings in their manuscript fully available?

Reviewer #1: Yes

Reviewer #2: Yes

5. Is the manuscript presented in an intelligible fashion and written in standard English?

Reviewer #1: Yes

Reviewer #2: Yes

6. Review Comments to the Author

Reviewer #1: (No Response)

Reviewer #2: The authors have critically addressed all the comments. The manuscript is acceptable for publication.

7. PLOS authors have the option to publish the peer review history of their article (what does this mean?). If published, this will include your full peer review and any attached files.

Reviewer #1: No

Reviewer #2: No

---

## [Editor Report · Acceptance letter]

1 Mar 2024

PONE-D-23-18137R1 

PLOS ONE

Dear Dr. Kaumba, 

I'm pleased to inform you that your manuscript has been deemed suitable for publication in PLOS ONE. Congratulations! Your manuscript is now being handed over to our production team.

Kind regards, 

on behalf of

Dr. Khin Thet Wai 

Academic Editor

PLOS ONE